# INDUCING GRAMMARS WITH AND FOR NEURAL MACHINE TRANSLATION

## ABSTRACT

Previous work has demonstrated the benefits of incorporating additional linguistic annotations such as syntactic trees into neural machine translation. However the cost of obtaining those syntactic annotations is expensive for many languages and the quality of unsupervised learning linguistic structures is too poor to be helpful. In this work, we aim to improve neural machine translation via source side dependency syntax but without explicit annotation. We propose a set of models that learn to induce dependency trees on the source side and learn to use that information on the target side. Importantly, we also show that our dependency trees capture important syntactic features of language and improve translation quality on two language pairs En-De and En-Ru.

## 1 INTRODUCTION

Sequence to sequence (seq2seq) models have exploded in popularity due to their apparent simplicity and yet surprising modeling strength. The basic architecture cleanly extends the standard machine learning paradigm wherein some function $f$ is learned to map inputs to outputs $x \rightarrow y$ to the case where $x$ and $y$ are natural language strings. In its most basic form, an input is summarized by a recurrent neural network into a summary vector and then decoded into a sequence of observations. These models have been strengthened with attention mechanisms (Bahdanau et al., 2015), and variational dropout (Gal & Ghahramani, 2016), in addition to important advances in expressivity via gating like Long Short-Term Memory (LSTM) cells (Hochreiter & Schmidhuber, 1997) and advanced gradient optimizers like Adam (Kingma & Ba, 2014).

Despite these impressive advances, the community has still largely been at a loss to explain how these models are so successful at a wide range of linguistic tasks. Recent work has shown that the LSTM captures a surprising amount of syntax (Linzen et al., 2016), but this is evaluated via downstream tasks designed to test the model's abilities not its representation.

Simultaneously, recent research in neural machine translation (NMT) has shown the benefit of modeling syntax explicitly using parse trees (Bastings et al., 2017; Li et al., 2017; Eriguchi et al., 2017) rather than assuming the model will automatically discover and encode it. Li et al. (2017) present a mixed encoding of words and a linearized constituency-based parse tree of the source sentence. Bastings et al. (2017) propose to use Graph Convolution to encode source sentences given their dependency links and attachment labels. In this work, we attempt to contribute to both modeling syntax and investigating a more interpretable interface for testing the syntactic content of a new seq2seq model's internal representation and attention.

We achieve this by augmenting seq2seq with a gate that allows the model to decide between syntactic and semantic objectives. The syntactic objective is encoded via a syntactic structured attention (Section §3) from which we can extract dependency trees. Our goal is to have a model which reaps the benefits of syntactic information (*i.e.* parse trees) without requiring explicit annotation. In this way, learning the internal representation of our model is a cousin to work done in unsupervised grammar induction except that by focusing on translation we require both syntactic and semantic knowledge. The semantic objective is the word translation prediction. It is often captured by attention, as an analogy to word-alignment model in phrase-based MT (Koehn et al., 2003). The syntactic objective is captured implicitly in the decoder because it ensures the fluency of the translation. For grammar induction, the translation objective is provides more guidance than the marginal likelihood

typically used in unsupervised learning. However, we note that the quality of the induced grammar also depends on the choice of the target language (§6).

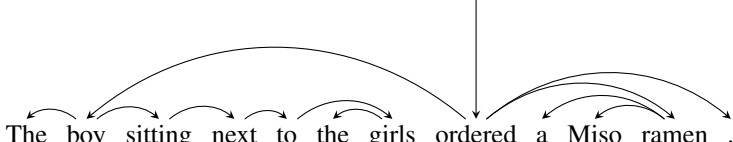

The boy sitting next to the girls ordered a Miso ramen .

Figure 1: Here we show a simple dependency tree. For the sake of understanding this paper, we draw the reader's eyes to two distinct classes of dependency: **semantic roles** (verb *ordered* → subject/object *boy, ramen*) and **syntactic rules** (noun *girls* → determiner *the*).

To motivate our work and the importance of structure in translation, consider the process of translating the sentence "*The boy sitting next to the girls ordered a Miso ramen.*" from English to German. The dependency tree of the sentence is given in figure 1. In German, translating the verb "*ordered*", requires knowledge of its subject "*boy*" to correctly predict the verb's number "*bestellte*" instead of "*bestellten*" if the model wrongly identifies "*girls*" as the subject. This is a case where syntactic agreement requires long-distance information transfer. On the other hand, translating the word "*next*" can be done in isolation without knowledge of neither its head nor child dependencies. While its true the decoder can, in principle, utilize previously predicted words (*e.g.* the translation of the "*boy*") to reason about subject-verb agreement, in practice LSTMs still struggle with long-distance dependencies. Moreover, Belinkov et al. (2017) showed that using attention reduces the capacity of the decoder to learn target side syntax.

Based on the insights from examples like the one above, we have designed a model with the following properties:

1. It can induce *syntactic* relations in the source sentences;
2. It can decide when and which syntactic information from the source to use for generating target words.

Previous work seems to imply that syntactic dependencies on the source side can be modeled via a self-attention layer (Vaswani et al., 2017) because self-attention allows direct interactions amongst source words. However, we will show that this is not always the case (section §6). We achieve our first requirement (1) by means of a syntactic attention layer (§3.1) that imposes non-projective dependency structure over the source sentence. To meet our second requirement (2) we use a simple gating mechanism (§3.2) that learns when to use the source side syntax.

As noted previously, in addition to demonstrating improvements in translation quality with our proposed models, we are also interested in analyzing the aforementioned non-projective dependency trees learned by the models. Recent work has begun analyzing task-specific latent trees (Williams et al., 2017). It has been shown that incorporating hierarchical structures leads to better task performance. Unlike the previous work that induced latent trees explicitly for semantic tasks, we present the first results on learning latent trees with a joint syntactic-semantic objective. We do this in the service of machine translation which inherently requires access to both aspects of a sentence.

In summary, in this work we make the following contributions:

- We propose a new NMT model that learns the latent structure of the encoder and how to use it during decoding. Our model is language independent and straightforward to apply with Byte-Pair Encoding (BPE) inputs. We show that our model obtains a significant improvement 0.6 BLEU (German→English) and 0.8 BLEU (English→German) over a strong baseline.
- We perform an in-depth analysis of the learned structures on the source side and investigate where the target decoder decides syntax is required.

The rest of the paper is organized as follow: We describe our NMT baseline in section §2. Our proposed models are detailed in section §3. We present the experimental setups and translation

results in section §4. In section §5 we analyze models' behavior by means of visualization which pairs with our analysis of the latent trees induced by our model in section §6. We conclude our work in the last section.

## 2 NEURAL MACHINE TRANSLATION

Given a training pair of source and target sentences $(\mathbf{x}, \mathbf{y})$ of length $n$ and $m$ respectively, Neural Machine Translation (NMT) is a conditional probabilistic model $p(\mathbf{y} \mid \mathbf{x})$ implemented using neural networks

$$\log p(\mathbf{y} \mid \mathbf{x};\, \boldsymbol{\theta}) = \sum_{j=1}^{m} \log p(\mathbf{y}_j \mid \mathbf{y}_{i<j}, \mathbf{x};\, \boldsymbol{\theta})$$

where $\boldsymbol{\theta}$ is the model's parameters. We will omit the parameters $\boldsymbol{\theta}$ herein for readability.

The NMT system used in this work is a seq2seq model that consists of a bidirectional LSTM encoder and an LSTM decoder coupled with an attention mechanism (Bahdanau et al., 2015; Luong et al., 2015). Our system is based on a PyTorch implementation[1] of OpenNMT (Klein et al., 2017). Let $\{\mathbf{s}_i \in \mathbb{R}^d\}_{i=1}^n$ be the output of the encoder

$$\mathbf{S} = \mathrm{enc}(\mathbf{x}) \tag{1}$$

Here we use $\mathbf{S} = [\mathbf{s}_1; \ldots; \mathbf{s}_n] \in \mathbb{R}^{d \times n}$ as a concatenation of $\{\mathbf{s}_i\}$. The decoder is composed of stacked LSTMs with input-feeding. Specifically, the inputs of the decoder at time step $t$ are the previous hidden state $\mathbf{h}_{t-1}$, a concatenation of the embedding of previous generated word $\mathbf{y}_{t-1}$ and a vector $\mathbf{u}_{t-1}$:

$$\mathbf{u}_{t-1} = g(\mathbf{h}_{t-1}, \mathbf{c}_{t-1}) \tag{2}$$

where $g$ is a one layer feed-forward network and $\mathbf{c}_{t-1}$ is a context vector computed by an attention mechanism

$$\boldsymbol{\alpha}_{t-1} = \mathrm{softmax}(\mathbf{h}_{t-1}^{\mathsf{T}} \mathbf{W}_a \mathbf{S}) \tag{3}$$

$$\mathbf{c}_{t-1} = \mathbf{S} \boldsymbol{\alpha}_{t-1}^{\mathsf{T}} \tag{4}$$

where $\mathbf{W}_a \in \mathbb{R}^{d \times d}$ is a trainable parameter.

Finally a single layer feed-forward network $f$ takes $\mathbf{u}_t$ as input and returns a multinomial distribution over all the target words

$$y_t \sim f(\mathbf{u}_t) \tag{5}$$

## 3 SYNTACTIC ATTENTION MODELS

Previous work on incorporating source-side syntax in NMT often focuses on modifying the standard recurrent encoder such that the encoder is explicitly made aware of the syntactic structure of the source sentence. Given a sentence of length $n$, syntax encoders of this type return a set of $n$ annotation vectors each compressing semantic and syntactic relations defined by the given parse tree of the input. The attention module then accesses these annotations during the generation of the target. We argue that this approach puts a lot of burden on the encoder as it has to balance the influence of semantics and syntax at every step regardless of the target words that are being generated. Here, we propose a simple alternative approach where we let the encoder output two sets of vectors: content annotations and syntactic annotations (Figure 2a). The content annotations are the outputs of a standard BiLSTM while the syntactic annotations are produced by a structured attention layer (§3.1). Having two set of annotations, first we compute attention weights $\boldsymbol{\alpha}$ using the decoder's hidden state $\mathbf{h}$, then we compute the context vector $\mathbf{c}$ (eq. 4) as in standard NMT system (Figure 2b). We then calculate syntactic vector $\mathbf{d}$ by taking a weighted average between $\boldsymbol{\alpha}$ and the syntactic annotations (Figure 2c). Finally, we allow the decoder to decide how much syntax it needs for making a prediction given the decoder's current state by using a gating mechanism to control syntactic information. Apart from lifting the burden otherwise placed on the encoder and tightly coupling the syntactic encoding to the

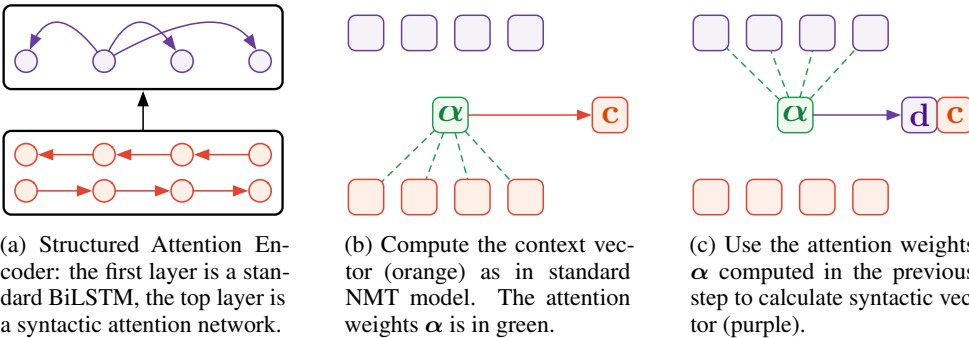

(a) Structured Attention Encoder: the first layer is a standard BiLSTM, the top layer is a syntactic attention network.

(b) Compute the context vector (orange) as in standard NMT model. The attention weights $\boldsymbol{\alpha}$ is in green.

(c) Use the attention weights $\boldsymbol{\alpha}$ computed in the previous step to calculate syntactic vector (purple).

Figure 2: Break down the computation of the proposed models.

decoder, the gating mechanism also allows us to inspect the decoder state and answer the question "*When does source side syntax matter?*" in section §5.

Inspired by structured attention networks (Kim et al., 2017), we present a syntactic attention layer that aims to discovery and convey source side dependency information to the decoder. The syntactic attention model consists of two parts:

1. A syntactic attention layer for head word selection in the encoder;

2. An attention with gating mechanism to control the amount of syntax needed for generating a target word at each time step.

### 3.1   HEAD WORD SELECTION

The head word selection layer learns to select a *soft* head word for each source word via structured attention. This layer does not have access to any dependency labels from the source. The head word selection layer transforms $\mathbf{S}$ into a matrix $\mathbf{M}$ that encodes implicit dependency structure of $\mathbf{x}$ using *self-structured-attention*. First we apply three trainable weight matrices $\mathbf{W}_q, \mathbf{W}_k, \mathbf{W}_v \in \mathbb{R}^{d \times d}$ to map $\mathbf{S}$ to query, key, and value matrices $\mathbf{S}_q, \mathbf{S}_k, \mathbf{S}_v \in \mathbb{R}^{d \times n}$:

$$\mathbf{S}_q = \mathbf{W}_q \mathbf{S} \qquad \mathbf{S}_k = \mathbf{W}_k \mathbf{S} \qquad \mathbf{S}_v = \mathbf{W}_v \mathbf{S} \tag{6}$$

Then we compute structured attention probabilities $\boldsymbol{\beta}$ relying on a function *sattn* that we will describe in detail shortly.

$$\boldsymbol{\beta} = \text{sattn}(\mathbf{S}_q^\mathsf{T} \mathbf{S}_k) \tag{7}$$

$$\mathbf{M} = \mathbf{S}_v \boldsymbol{\beta} \tag{8}$$

The structured attention function *sattn* is inspired by the work of Kim et al. (2017) but differs in two important ways. First we model *non-projective dependency trees*. Second, we ultilize Kirchhoff's Matrix-Tree Theorem (Tutte, 1984) instead of sum-product algorithm presented in (Kim et al., 2017) for fast evaluation of the attention probabilities. We note that Liu & Lapata (2017) first propose using the Matrix-Tree Theorem for evaluating the marginals in end to end training of neural networks. Their work however focuses on semantic objectives rather than a joint semantic and syntactic objectives such as machine translation. Additionally, in this work, we will evaluate structured attention component on datasets that are two orders of magnitude larger than the datasets studied in Liu & Lapata (2017).

Let $\mathbf{z} \in \{0, 1\}^{n \times n}$ be an adjacency matrix encoding a source's dependency tree. Let $\phi \in \mathbb{R}^{n \times n}$ be a scoring matrix such that cell $\phi_{i,j}$ scores how likely word $x_i$ is to be the head of word $x_j$. The matrix $\phi$ is obtained simply by

$$\phi = \mathbf{S}_q^\mathsf{T} \mathbf{S}_k \tag{9}$$

---

[1]http://opennmt.net/OpenNMT-py/

The probability of a dependency tree $\mathbf{z}$ is therefore given by

$$p(\mathbf{z} \,|\, \mathbf{x}; \boldsymbol{\phi}) = \frac{\exp\left(\sum_{i,j} \mathbf{z}_{i,j} \, \boldsymbol{\phi}_{i,j}\right)}{Z(\boldsymbol{\phi})} \tag{10}$$

where $Z(\boldsymbol{\phi})$ is the partition function.

In the head selection model, we are interested in the marginal $p(\mathbf{z}_{i,j} = 1 \,|\, \mathbf{x}; \boldsymbol{\phi})$

$$\boldsymbol{\beta}_{i,j} = p(\mathbf{z}_{i,j} = 1 \,|\, \mathbf{x}; \boldsymbol{\phi}) = \sum_{\mathbf{z}\,:\,\mathbf{z}_{i,j}=1} p(\mathbf{z} \,|\, \mathbf{x}; \boldsymbol{\phi}) \tag{11}$$

We use the framework presented by Koo et al. (2007) to compute the marginal of non-projective dependency structures. Koo et al. (2007) use the Kirchhoff's Matrix-Tree Theorem (Tutte, 1984) to compute $p(\mathbf{z}_{i,j} = 1 \,|\, \mathbf{x}; \boldsymbol{\phi})$ as follow:

$$\mathbf{L}_{i,j}(\boldsymbol{\phi}) = \begin{cases} \sum\limits_{\substack{k=1 \\ k \neq j}}^{n} \exp(\boldsymbol{\phi}_{k,j}) & \text{if } i = j \\ -\exp(\boldsymbol{\phi}_{i,j}) & \text{otherwise} \end{cases} \tag{12}$$

Now we construct a matrix $\hat{\mathbf{L}}$ that accounts for root selection

$$\hat{\mathbf{L}}_{i,j}(\boldsymbol{\phi}) = \begin{cases} \exp(\boldsymbol{\phi}_{j,j}) & \text{if } i = 1 \\ \mathbf{L}_{i,j}(\boldsymbol{\phi}) & \text{if } i > 1 \end{cases} \tag{13}$$

The marginals $\boldsymbol{\beta}$ are then

$$\boldsymbol{\beta}_{i,j} = (1 - \delta_{1,j}) \exp(\boldsymbol{\phi}_{i,j}) \left[\hat{\mathbf{L}}^{-1}(\boldsymbol{\phi})\right]_{j,j} - (1 - \delta_{i,1}) \exp(\boldsymbol{\phi}_{i,j}) \left[\hat{\mathbf{L}}^{-1}(\boldsymbol{\phi})\right]_{j,i} \tag{14}$$

where $\delta_{i,j}$ is the Kronecker delta. For the root node, the marginals are given by

$$\boldsymbol{\beta}_{k,k} = \exp(\boldsymbol{\phi}_{k,k}) \left[\hat{\mathbf{L}}^{-1}(\boldsymbol{\phi})\right]_{k,1} \tag{15}$$

The computation of the marginals is fully differentiable, thus we can train the model in an end-to-end fashion by maximizing the conditional likelihood of the translation.

## 3.2 Incorporating syntactic context

We encourage the decoder to use syntactic annotations by means of attention. Essentially, if the model attends to a particular source word $x_i$ when generating the next target word, we also want the model to attend to the head word of $x_i$. We implement this idea using a new *shared attention* layer from decoder's state $\mathbf{h}$ to encoder's annotations $\mathbf{S}$ and $\mathbf{M}$. First, a we compute a standard attention weights $\boldsymbol{\alpha}_{t-1} = \text{softmax}(\mathbf{h}_{t-1}^{\top} \mathbf{W}_a \mathbf{S})$ as in equation 3. We then compute a weighted syntactic vector:

$$\mathbf{d}_{t-1} = \mathbf{M} \boldsymbol{\alpha}_{t-1}^{\top} \tag{16}$$

Note that the syntactic vector $\mathbf{d}_{t-1}$ and the context vector $\mathbf{c}_{t-1}$ share the same attention weights $\boldsymbol{\alpha}_{t-1}$ at time step $t$. By sharing the attention weights $\boldsymbol{\alpha}_{t-1}$ we hope that if the model picks a source word $x_i$ to translate with the highest probability $\boldsymbol{\alpha}_{t-1}[i]$, the contribution of $x_i$'s head in the syntactic vector $\mathbf{d}_{t-1}$ is also highest. It is not always useful or necessary to access the syntactic context $\mathbf{d}_{t-1}$ every time step $t$. Ideally, we should let the model decide whether it needs to use this information. For example, the model might decide when it needs to resolve long distance dependencies in the source side. To control the amount of source side syntactic information we introduce a gating mechanism:

$$\hat{\mathbf{d}}_{t-1} = \mathbf{d}_{t-1} \odot \sigma(\mathbf{W}_g \mathbf{h}_{t-1}) \tag{17}$$

The vector $\mathbf{u}_{t-1}$ from equation 2 now becomes

$$\mathbf{u}_{t-1} = g(\mathbf{h}_{t-1}, \mathbf{c}_{t-1}, \hat{\mathbf{d}}_{t-1}) \tag{18}$$

| | | | | | |
|---|---|---|---|---|---|
| $\boldsymbol{\alpha}$ 0.01 | 0.02 | 0.02 | 0.85 | 0.02 | 0.05 |
| boy | orders | to | boy | ramen | orders |
| | | | | | |
| The | boy ... | girls | ordered ... | miso | ramen |
| $\boldsymbol{\alpha}$ 0.01 | 0.02 | 0.02 | 0.85 | 0.02 | 0.05 |

(a) shared attention.

| | | | | | |
|---|---|---|---|---|---|
| $\boldsymbol{\gamma}$ 0.03 | 0.35 | 0.01 | 0.05 | 0.01 | 0.40 |
| boy | orders | to | boy | ramen | orders |
| | | | | | |
| The | boy ... | girls | ordered ... | miso | ramen |
| $\boldsymbol{\alpha}$ 0.01 | 0.02 | 0.02 | 0.85 | 0.02 | 0.05 |

(b) having two separate attentions.

Figure 3: A pictorial illustration of having two separate attention (3b) and shared attention (3a) from the decoder to the encoder. The blue text represents the content vectors of the sentence and the purple text represents the syntactic vectors. The number corresponding to each word is the probability mass from decoder-to-encoder attention layer(s). Note, the reallocation of mass to both the subject and object.

An alternative to incorporate syntactic annotation $\mathbf{M}$ to the decoder is to use a separate attention layer to compute the syntactic vector $\mathbf{d}_{t-1}$ at time step $t$:

$$\boldsymbol{\gamma}_{t-1} = \text{softmax}(\mathbf{h}_{t-1}^{\mathsf{T}}\mathbf{W}_m\mathbf{M}) \tag{19}$$

$$\mathbf{d}_{t-1} = \mathbf{M}\boldsymbol{\gamma}_{t-1}^{\mathsf{T}} \tag{20}$$

Figure 3 illustrates the difference between shared attention and separate attention when the decoder is translating the english word "ordered". The source words are in blue and their corresponding head words are in purple. As can be seen, shared attention now helps the decoder pick the right number for the verb by taking into account the subject "boy".

### 3.3 HARD ATTENTION OVER TREE STRUCTURES

Finally, we include an experiment with hard structured attention. The main motivation of this experiment is twofold. First, we want to simulate the scenario where the model has access to a decoded parse tree. Obviously, we do not expect this model to perform best overall in NMT as it only has access to an induced tree rather than a gold one. Conversely, forcing the model to make hard decisions during training mirrors the intermediary extraction and conditioning on a dependency tree (§6.1), we therefore hope this technique will improve the performance on grammar induction.

Recall the marginal $\boldsymbol{\beta}_{i,j}$ gives us the probability that word $x_i$ is the head of word $x_j$. We convert these soft weights to hard ones $\bar{\boldsymbol{\beta}}$ by

$$\bar{\boldsymbol{\beta}}_{k,j} = \begin{cases} 1 & \text{if } k = \arg\max_i \boldsymbol{\beta}_{i,j} \\ 0 & \text{otherwise} \end{cases} \tag{21}$$

We train this model using the straight-through estimator (Bengio et al., 2013). Note that in this setup, each word has a parent but there is no guarantee that the structure given by hard attention will result in a tree (*i.e.* it may contain cycle). A more principle way to enforce tree structure is to decode the best tree $\mathcal{T}$ using the maximum spanning tree algorithm (Chu & Liu, 1965; Edmonds, 1967) and to set $\bar{\boldsymbol{\beta}}_{k,j} = 1$ if the edge $(x_k \to x_j) \in \mathcal{T}$. Unfortunately, maximum spanning tree decoding can be prohibitively slow as the Chu-Liu-Edmonds algorithm is not GPU friendly. We therefore resort to greedily picking a parent word for each word $x_j$ in the sentence using equation 21. This is actually a principled simplification as greedily assigning a parent for each word is the first step in Chu-Liu-Edmonds algorithm.

## 4 EXPERIMENTS

Next we will discuss our experimental setup and report results for English↔German (En↔De) and English↔Russian (Ru↔En) translation models.

## 4.1 DATA

We use WMT17[2] data in our experiments. Table 1 shows the statistics of the data. For En–De, we use a concatenation of Europarl, Common Crawl, Rapid corpus of EU press releases, and News Commentary v12. We use *newstest2015* for development and *newstest2016*, *newstest2017* for test. For En–Ru, we use Common Crawl, News Commentary v12, and Yandex Corpus. The development data comes from *newstest2016* and *newstest2017* and is reserved for testing.

Table 1: Statistics of the data used in our experiment.

|  | Train | Valid | Test | | Vocabulary | |
|---|---|---|---|---|---|---|
|  |  |  | wmt16 | wmt17 | En | Other |
| En–De | 5.9M | 2,169 | 2,999 | 3,004 | 36,251 | 35,913 |
| En–Ru | 2.1M | 2,998 | – | 3,001 | 34,872 | 34,989 |

We use BPE (Sennrich et al., 2016) with 32,000 merge operations. We run BPE for each language instead of using BPE for the concatenation of both source and target languages.

## 4.2 BASELINES

Our baseline is an NMT model with input-feeding (§2). As we will be making several modifications from the basic architecture in our proposed models, we will verify each choice in our architecture design empirically. First we validate the structured attention module by comparing it to a self-attention module (Lin et al., 2017; Vaswani et al., 2017). Since self-attention does not assume any hierarchical structure over the source sentence, we refer it as flat-attention (FA). Second, we validate the benefit of using two sets of annotations in the encoder. We combine the hidden states of the encoder $\mathbf{h}$ with syntactic context $\mathbf{d}$ to obtain a single set of annotation using the following equation

$$\bar{\mathbf{s}}_i = \mathbf{s}_i + \sigma(\mathbf{W}_g \mathbf{s}_i) \odot \mathbf{d}_i \tag{22}$$

Here we first down weight the syntactic context $\mathbf{d}_i$ before adding it to $\mathbf{s}_i$. We refer to this baseline as SA-NMT-1set. Note that in this baseline, there is only one attention layer from the target to the source.

In all the models, we share the weights of target word embeddings and the output layer as suggested by Inan et al. (2017); Press & Wolf (2017).

## 4.3 HYPER-PARAMETERS AND TRAINING

For all the models, we set the word embedding size to 1024, the number of LSTM layers to 2, and the dropout rate to 0.3. Parameters are initialized uniformly in $(-0.04, 0.04)$. We use the Adam optimizer with an initial learning rate 0.001. We evaluate our models on development data every 10,000 updates for De-En and 5,000 updates for Ru-En. If the validation perplexity increases, we decay the learning rate by 0.5. We stop training after decaying the learning rate five times as suggested by Denkowski & Neubig (2017). The mini-batch size is 32 in all the experiments. We report the BLEU scores using the `multi-bleu.perl` script.

## 4.4 RESULTS

Table 2 shows the BLEU scores in our experiments. We test statistical significance using bootstrap resampling (Riezler & Maxwell, 2005). Statisical significance are marked as $^{\dagger}p < 0.05$ and $^{\ddagger}p < 0.01$ when compared against the baselines. Additionally, we also report statistical significance $^{\triangle}p < 0.05$ and $^{\blacktriangle}p < 0.01$ when compared against the FA-NMT models that have two separate attention layers from the decoder to the encoder. Overall, the SA-NMT (shared) model performs the best gaining more than 0.5 BLEU De→En on wmt16, up to 0.82 BLEU on En→De wmt17 and 0.64 BLEU En→Ru direction over a competitive NMT baseline. The results show that structured attention is useful when translating from English to languages that have long-distance dependencies

---

[2]http://www.statmt.org/wmt17

and complex morphological agreements. We also see that the gain is marginal compared to self-attention models (shared-FA-NMT-shared) and not significant. Within FA-NMT models, sharing attention is helpful. Our results also confirm the advantage of having two separate sets of annotations in the encoder when modeling syntax. The hard structured attention model (SA-NMT-hard) performs comparable to the baseline. While this is a somewhat expected result from the hard attention model, we will show in the next section (§6) that the quality of induced trees from hard attention is far better than the soft ones.

Table 2: Results for translating En-De and En-Ru both directions. In all of our experiments, while the SA-NMT-shared model does not statistically outperform FA-NMT-shared it does outperform FA-NMT with separate attentions in three benchmarks. The results show that our proposed *shared-attention* is a benefit for NMT.

| Model | Shared | De→En | | Ru→En | En→De | | En→Ru |
| --- | --- | --- | --- | --- | --- | --- | --- |
| | | wmt16 | wmt17 | wmt17 | wmt16 | wmt17 | wmt17 |
| NMT | - | 33.16 | 28.94 | 30.17 | 29.92 | 23.44 | 26.41 |
| FA-NMT | yes | 33.55 | 29.43 | 30.22 | 30.09 | 24.03 | 26.91 |
| | no | 33.24 | 29.00 | 30.34 | 29.98 | 23.97 | 26.75 |
| SA-NMT-1set | - | 33.51 | 29.15 | 30.34 | **30.29**$^\dagger$ | 24.12 | 26.96 |
| SA-NMT-hard | yes | 33.38 | 28.96 | 29.98 | 29.93 | 23.84 | 26.71 |
| SA-NMT | yes | **33.73**$^{\ddagger\triangle}$ | **29.45**$^{\ddagger\blacktriangle}$ | **30.41** | 30.22 | **24.26**$^{\ddagger\triangle}$ | **27.05**$^\ddagger$ |
| | no | 33.18 | 29.19 | 30.15 | 30.17 | 23.94 | 27.01 |

## 5 ATTENTION AND GATE ACTIVATION VISUALIZATION

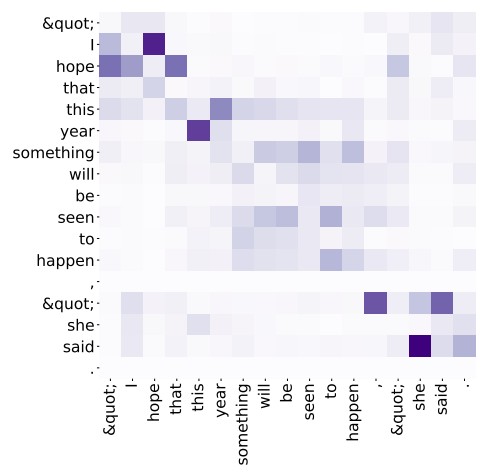
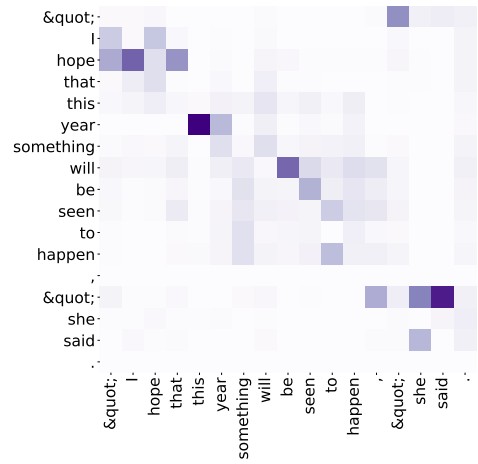

(a) SA-NMT (shared) attention.          (b) SA-NMT with hard structured attention.

Figure 4: A visualization of attention distributions over head words. y-axis shows the head words. Darker color means higher attention weights. As can be seen here, while both models agree on some basic elements of the underlying grammar, the attention's mass tends to concentrate on fewer tokens in hard structured attention. For some tokens, hard-attention, before binarization by eq 21, does not show a strong favor towards any head. Perhaps this explains the poor performance of SA-NMT-hard in translation because hard attention has to pick one head word among all equally probable heads.

Figure 4 shows a sample visualization of structured attention models trained on En→De data. It is worth noting that the shared SA-NMT model (Figure 4a) and the hard SA-NMT model (Figure 4b) capture similar structures of the source sentence. We hypothesize that when the objective function requires syntax, the induced trees are more consistent unlike those discovered by a semantic objective (Williams et al., 2017). Both models correctly identify that the verb is the head of pronoun (hope→I,

said→she). While intuitively it is clearly beneficial to know the subject of the verb when translating from English into German, the model attention is still somewhat surprising because long distance dependency phenomena are less common in English, so we would expect that a simple content based addressing (*i.e.* standard attention mechanism) would be sufficient in this translation. Finally, in addition to attention weight visualization, we provide sample trees induced by our models in Figure 5.

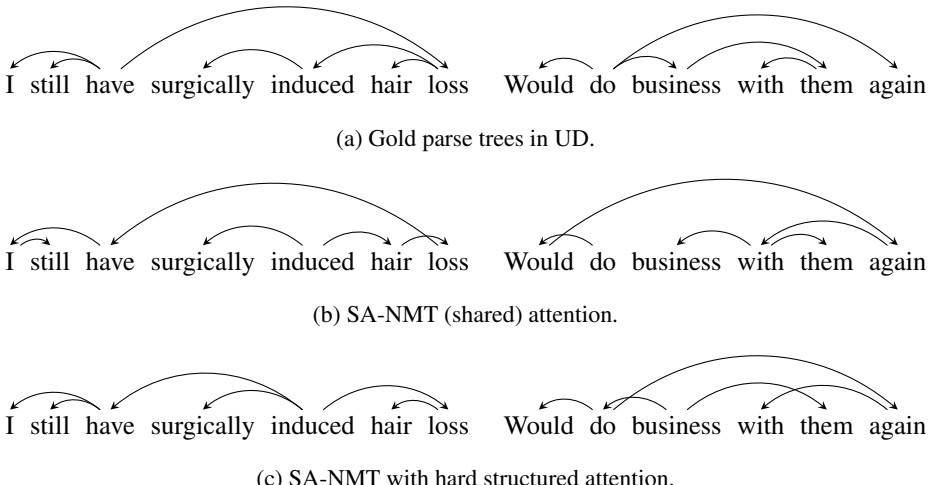

(a) Gold parse trees in UD.

(b) SA-NMT (shared) attention.

(c) SA-NMT with hard structured attention.

Figure 5: Trees induced by the SA-NMT models. Roots and punctuations are ingored.

We now turn to the question of when does the target LSTM need to access source side syntax. We investigate this by analyzing the gate activations of our best model, SA-NMT (shared). At time step $t$, when the model is about to predict the target word $y_t$, we compute the norm of the gate activations

$$z_t = \|\sigma(\mathbf{W}_g \mathbf{h}_{t-1})\|_2 \qquad (23)$$

The activation norm $z_t$ allows us to see how much syntactic information flows into the decoder. We observe that $z_t$ has its highest value when the decoder is about to generate a verb while it has its lowest value when the end of sentence token `` is predicted. Figure 6 shows some examples of German target sentences. The darker colors represent higher activation norms and bold words indicate the highest activation norms when those words are being predicted.

Figure 6: Visualization of gate norm. Best viewed in color.

Es ist schon ko@@ misch , wie dies immer wieder zu dieser Jahreszeit auf@@ taucht .

Das braucht Zeit und Mut .

Das Dach darf für Solar@@ anlage fla@@ cher werden .

Hunderte Flüchtlinge sollen in Wies@@ baden unter@@ kommen

Am Mitt@@ wo@@ ch@@ morgen hätten sich dort noch 40 Flüchtlinge aufge@@ halten .

Oder wollen Sie herausfinden , über was andere reden ?

It is clear that translating verbs requires knowledge of syntax. We also see that after verbs, the gate activation norms are highest at nouns Zeit (*time*), Mut (*courage*), Dach (*roof*) and then tail off as we move to function words which require less context to disambiguate. Below are the frequencies with which the highest activation norm in a sentence is applied to a given part-of-speech tag on *newstest2016*. We include the top 10 most common activations. It is important to note that this distribution is dramatically different than a simple frequency baseline.

| VERB | NOUN | AUX | ADP | PUNCT | ADJ | DET | PART | PROPN | ADV |
|---|---|---|---|---|---|---|---|---|---|
| 1022 | 636 | 193 | 189 | 184 | 170 | 167 | 95 | 75 | 71 |

# 6  GRAMMAR INDUCTION

NLP has longed assumed hierarchical structured representations were important to understanding language. In this work, we have borrowed that intuition to inform the construction of our model (as previously discussed). We feel it is important to take a step beyond a comparison of aggregate model performance and investigate whether the internal latent representations discovered by our models share properties previously identified within linguistics and if not, what important differences exist. We investigate the interpretability of our model's representations by: 1) A quantitative attachment accuracy and 2) A qualitative comparison of the underlying grammars.

Our results both corroborate and refute previous work (Hashimoto & Tsuruoka, 2017; Williams et al., 2017). We agree and provide stronger evidence that syntactic information can be discovered via latent structured attention, but we also present preliminary results that indicate that conventional definitions of syntax may be at odds with task specific performance.

## 6.1  EXTRACTING A TREE

For extracting non-projective dependency trees, we use Chu-Liu-Edmonds algorithm (Chu & Liu, 1965; Edmonds, 1967). First, we must collapse BPE segments into words. Assume the $k$-th word corresponds to BPE tokens from index $u$ to $v$. We obtain a new matrix $\hat{\phi}$ by summing over $\phi_{i,j}$ that are the corresponding BPE segments.

$$\hat{\phi}_{i,j} = \begin{cases} \phi_{i,j} & \text{if } i \notin [u,v] \land j \notin [u,v] \\ \sum_{l=u}^{v} \phi_{i,l} & \text{if } j = k \land i \notin [u,v] \\ \sum_{l=u}^{v} \phi_{l,j} & \text{if } i = k \land j \notin [u,v] \\ \sum_{l=u}^{v} \sum_{h=u}^{v} \phi_{l,h} & \text{otherwise} \end{cases} \tag{24}$$

## 6.2  GRAMMATICAL ANALYSIS

We compute unlabeled directed and undirected attachment accuracies of our predicted trees on gold annotations from Universal Dependencies (UD version 2) dataset[3]. Our five model settings in addition to left and right branching baselines are presented in Table 3. The results indicate that the target language effects the source encoder's induction performance and several settings are competitive with branching baselines for determining headedness. We see performance gains from hard attention and several models outperform baselines for undirected dependency metrics (UA). Whether hard attention helps is unclear. Its appears to help for German and not with Russian.

Successfully extracting linguistic structure with hard attention indicates that models can capture interesting structures beyond semantic co-occurrence via discrete actions. This corroborates previous work (Choi et al., 2017; Yogatama et al., 2017) which has shown that non-trivial structures are learned by using REINFORCE (Williams, 1992) or Gumbel-softmax trick (Jang et al., 2016) to backprop through discrete units. Our approach also outperforms that of Hashimoto & Tsuruoka (2017) despite our model lacking access to additional resources like part-of-speech tags.

**Dependency Accuracies**  While SA-NMT-hard model gives the best directed attachment scores on both German and English, the BLEU scores of this model are below other SA-NMT models as shown in Table 2. The lack of correlation between syntactic performance and NMT contradicts the intuition of previous work and actually suggests that useful structures learned in service of a task might not necessarily benefit from or correspond to known linguistic formalisms.

---

[3] http://universaldependencies.org

Table 3: Directed and Undirected (DA/UA) accuracies of our models on both English and German data as compared to branching baselines. Punctuation is removed during the evaluation. Our results show an intriguing effect of the target language on grammar induction. We observe a huge boost in DA/UA scores in FA-NMT and SA-NMT-shared models when the target language is morphologically rich (Russian). In comparison to previous work (Belinkov et al., 2017; Shi et al., 2016) on the encoder's ability to capture source side syntax, we show a stronger result that even when the encoders are designed to capture syntax explicitly, the choice of the target language has a great influence on the amount of syntax learned by the encoder.

| | FA | | SA | | | Baseline | |
|---|---|---|---|---|---|---|---|
| | no-shared | shared | no-shared | shared | hard | L | R |
| EN (→de) | 17.0/25.2 | 27.6/41.3 | 23.6/33.7 | 27.8/42.6 | **31.7/45.6** | 34.0/40.5 | 7.8/40.9 |
| EN (→ru) | 35.2/48.5 | **36.5**/48.8 | 12.8/25.5 | 33.1/**48.9** | 33.7/46.0 | | |
| DE (→en) | 21.1/33.3 | 20.1/33.6 | 12.8/22.5 | 21.5/38.0 | **26.3/40.7** | 34.4/42.8 | 8.6/41.5 |
| RU (→en) | 23.2/38.1 | 26.3/43.0 | 21.8/37.5 | **26.5/44.3** | 22.5/36.6 | 32.9/47.3 | 15.2/47.3 |

Table 4: Most common grammar rules and their production percentages in EN and DE. English's strict left branching structure makes it difficult to outperform, but we see substantial gains by our approach on the more syntactic elements of language (*e.g.* DET/ADJ/ADP attachments). For EN, we use en→ru systems.

| | | EN | | | | DE | | | |
|---|---|---|---|---|---|---|---|---|---|
| | | gold | left | SA | hard | gold | left | SA | hard |
| VERB | → NOUN | 25.1 | 9.5 | 7.5 | **9.8** | 34.7 | **41.7** | 6.2 | 9.0 |
| | → PRON | 18.6 | **24.9** | 25.9 | 25.7 | 14.1 | 15.7 | **14.0** | 15.3 |
| | → ADV | 9.1 | **9.9** | 10.5 | 11.7 | 11.6 | 9.8 | 9.9 | **10.3** |
| | → VERB | 12.8 | **5.4** | 3.2 | 4.5 | 6.6 | 2.4 | 1.6 | **3.2** |
| NOUN | → DET | 23.2 | 19.7 | 20.1 | **24.6** | 27.2 | 39.5 | 17.6 | **19.0** |
| | → ADP | 17.2 | 14.3 | **17.7** | 16.6 | 17.3 | 8.8 | **12.9** | 11.4 |
| | → NOUN | 18.4 | 20.4 | 14.1 | **17.4** | 15.7 | 4.6 | **15.3** | 17.4 |
| | → ADJ | 13.9 | **13.7** | 14.6 | 16.0 | 13.9 | 25.5 | **16.1** | 19.0 |

**Qualitative Grammar Analysis**   We should obviously note that the model's strength shows up in the directed but not the undirected attention. This begs the question as to whether there are basic structural elements the grammar has decided not to attend to or if all constructions are just generally weak. We qualitatively analyzed the learned grammars as a function of dependency productions between universal part-of-speech tags in Table 4. Here, we extract the grammar of the language as if it were a CFG and compare the gold production frequencies and compare them to our models' predictions. In other words, how often does $\text{tag}_i$ generate $\text{tag}_j$ in the treebank and how closely did our models uncover those statistics. This finer grained analysis gives us insight into the model's surprising ability to often verb based syntax when translating, but simultaneously favoring noun based constructions. This is particularly noticeable in the SA model for German.

# 7   CONCLUSION

We have proposed a structured attention encoder for NMT. Our models show significant gains in performance over a strong baseline on standard WMT benchmarks. The models presented here do not access any external information such as parse-trees or part-of-speech tags. We show that our models induce dependency trees over the source sentences that systematically outperform baseline branching and previous work. We find that the quality of induced trees (compared against gold standard annotations) is not correlated with the translation quality.

R EFERENCES

Dzmitry Bahdanau, Kyunghyun Cho, and Yoshua Bengio. Neural machine translation by jointly learning to align and translate. In *ICLR 2015*, San Diego, CA, USA, May 2015.

Joost Bastings, Ivan Titov, Wilker Aziz, Diego Marcheggiani, and Khalil Simaan. Graph convolutional encoders for syntax-aware neural machine translation. In *Proceedings of the 2017 Conference on Empirical Methods in Natural Language Processing*, pp. 1947–1957. Association for Computational Linguistics, 2017.

Yonatan Belinkov, Nadir Durrani, Fahim Dalvi, Hassan Sajjad, and James Glass. What do neural machine translation models learn about morphology? In *Proceedings of the 55th Annual Meeting of the Association for Computational Linguistics (Volume 1: Long Papers)*, pp. 861–872. Association for Computational Linguistics, 2017. doi: 10.18653/v1/P17-1080.

Yoshua Bengio, Nicholas Léonard, and Aaron Courville. Estimating or propagating gradients through stochastic neurons for conditional computation. *arXiv preprint arXiv:1308.3432*, 2013.

Jihun Choi, Kang Min Yoo, and Sang goo Lee. Learning to compose task-specific tree structures. *AAAI*, 2017.

Y. J. Chu and T. H. Liu. On the shortest arborescence of a directed graph. *Science Sinica*, 14, 1965.

Michael Denkowski and Graham Neubig. Stronger baselines for trustable results in neural machine translation. In *Proceedings of the First Workshop on Neural Machine Translation*, pp. 18–27, Vancouver, August 2017. Association for Computational Linguistics.

Jack Edmonds. Optimum Branchings. *Journal of Research of the National Bureau of Standards*, 71B:233–240, 1967.

Akiko Eriguchi, Yoshimasa Tsuruoka, and Kyunghyun Cho. Learning to parse and translate improves neural machine translation. In *Proceedings of the 55th Annual Meeting of the Association for Computational Linguistics (Volume 2: Short Papers)*, pp. 72–78. Association for Computational Linguistics, 2017. doi: 10.18653/v1/P17-2012.

Yarin Gal and Zoubin Ghahramani. A theoretically grounded application of dropout in recurrent neural networks. In *NIPS*, 2016.

Kazuma Hashimoto and Yoshimasa Tsuruoka. Neural machine translation with source-side latent graph parsing. In *Proceedings of the 2017 Conference on Empirical Methods in Natural Language Processing*, pp. 125–135. Association for Computational Linguistics, 2017.

Sepp Hochreiter and Jürgen Schmidhuber. Long short-term memory. *Neural computation*, 9(8): 1735–1780, 1997.

Hakan Inan, Khashayar Khosravi, and Richard Socher. Tying word vectors and word classifiers: A loss framework for language modeling. *ICLR*, 2017.

Eric Jang, Shixiang Gu, and Ben Poole. Categorical reparameterization with gumbel-softmax. In *International Conference on Learning Representations*, 2016.

Yoon Kim, Carl Denton, Luong Hoang, and Alexander M. Rush. Structured attention networks. *ICLR*, 2017.

Diederik P. Kingma and Jimmy Ba. Adam: A method for stochastic optimization. In *International Conference on Learning Representations (ICLR)*, 2014.

Guillaume Klein, Yoon Kim, Yuntian Deng, Jean Senellart, and Alexander Rush. Opennmt: Open-source toolkit for neural machine translation. In *Proceedings of ACL 2017, System Demonstrations*, pp. 67–72, Vancouver, Canada, July 2017. Association for Computational Linguistics.

Philipp Koehn, Franz Josef Och, and Daniel Marcu. Statistical phrase-based translation. In *Proceedings of HLT-NAACL 2003*, pp. 127–133, Edmonton, Canada, 2003.

Terry Koo, Amir Globerson, Xavier Carreras, and Michael Collins. Structured prediction models via the matrix-tree theorem. In *Proceedings of the 2007 Joint Conference on Empirical Methods in Natural Language Processing and Computational Natural Language Learning (EMNLP-CoNLL)*, pp. 141–150, Prague, Czech Republic, June 2007. Association for Computational Linguistics.

Junhui Li, Deyi Xiong, Zhaopeng Tu, Muhua Zhu, Min Zhang, and Guodong Zhou. Modeling source syntax for neural machine translation. In *Proceedings of the 55th Annual Meeting of the Association for Computational Linguistics (Volume 1: Long Papers)*, pp. 688–697. Association for Computational Linguistics, 2017. doi: 10.18653/v1/P17-1064.

Zhouhan Lin, Minwei Feng, Cicero Nogueira dos Santos, Mo Yu, Bing Xiang, Bowen Zhou, and Yoshua Bengio. A Structured Self-attentive Sentence Embedding. *ArXiv e-prints*, March 2017.

Tal Linzen, Emmanuel Dupoux, and Yoav Goldberg. Assessing the ability of lstms to learn syntax-sensitive dependencies. *Transactions of the Association for Computational Linguistics*, 4:521–535, 2016. ISSN 2307-387X.

Yang Liu and Mirella Lapata. Learning structured text representations. *CoRR*, abs/1705.09207, 2017.

Thang Luong, Hieu Pham, and Christopher D. Manning. Effective approaches to attention-based neural machine translation. In *Proceedings of the 2015 Conference on Empirical Methods in Natural Language Processing*, pp. 1412–1421, Lisbon, Portugal, September 2015. Association for Computational Linguistics.

Ofir Press and Lior Wolf. Using the output embedding to improve language models. In *Proceedings of the 15th Conference of the European Chapter of the Association for Computational Linguistics: Volume 2, Short Papers*, pp. 157–163. Association for Computational Linguistics, 2017.

Stefan Riezler and John T. Maxwell. On some pitfalls in automatic evaluation and significance testing for mt. In *Proceedings of the ACL Workshop on Intrinsic and Extrinsic Evaluation Measures for Machine Translation and/or Summarization*, pp. 57–64, Ann Arbor, Michigan, June 2005. Association for Computational Linguistics.

Rico Sennrich, Barry Haddow, and Alexandra Birch. Neural machine translation of rare words with subword units. In *Proceedings of the 54th Annual Meeting of the Association for Computational Linguistics (Volume 1: Long Papers)*, pp. 1715–1725, Berlin, Germany, August 2016. Association for Computational Linguistics.

Xing Shi, Inkit Padhi, and Kevin Knight. Does string-based neural mt learn source syntax? In *Proceedings of the 2016 Conference on Empirical Methods in Natural Language Processing*, pp. 1526–1534, Austin, Texas, November 2016. Association for Computational Linguistics.

W. T Tutte. *Graph theory*. Cambridge University Press, 1984.

Ashish Vaswani, Noam Shazeer, Niki Parmar, Jakob Uszkoreit, Llion Jones, Aidan N Gomez, Ł ukasz Kaiser, and Illia Polosukhin. Attention is all you need. In I. Guyon, U. V. Luxburg, S. Bengio, H. Wallach, R. Fergus, S. Vishwanathan, and R. Garnett (eds.), *Advances in Neural Information Processing Systems 30*, pp. 6000–6010. Curran Associates, Inc., 2017.

Adina Williams, Andrew Drozdov, and Samuel R. Bowman. Learning to parse from a semantic objective: It works. is it syntax? *ArXiv e-prints*, September 2017.

Ronald J. Williams. Simple statistical gradient-following algorithms for connectionist reinforcement learning. *Machine Learning*, 8(3-4):229–256, May 1992. ISSN 0885-6125. doi: 10.1007/BF00992696.

Dani Yogatama, Phil Blunsom, Chris Dyer, Edward Grefenstette, and Wang Ling. Learning to compose words into sentences with reinforcement learning. *International Conference on Learning Representations*, 2017.

