# OpenReview forum: "Inducing Grammars with and for Neural Machine Translation"
_ICLR.cc/2018/Conference — Reject_

### Official Review · AnonReviewer3 · 2017-11-23

**Rating:** 3
**Confidence:** 5

**Review:**

This paper adds source side dependency syntax trees to an NMT model without explicit supervision. Exploring the use of syntax in neural translation is interesting but I am not convinced that this approach actually works based on the experimental results.

The paper distinguishes between syntactic and semantic objectives (4th paragraph in section 1), attention, and heads. Please define what semantic attention is. You just introduce this concept without any explanation. I believe you mean standard attention, if so, please explain why standard attention is semantic.

Clarity. What is shared attention exactly? Section 3.2 says that you share attention weights from the decoder with encoder. Please explain this a bit more. Also the example in Figure 3 is not very clear and did not help me in understanding this concept.

Results. A good baseline would be to have two identical attention mechanisms to figure out if improvements come from more capacity or better model structure. Flat attention seems to add a self-attention model and is somewhat comparable to two mechanisms. The results show hardly any improvement over the flat attention baseline (at most 0.2 BLEU which is well within the variation of different random initializations). It looks as if the improvement comes from adding additional capacity to the model.

Equation 3: please define H.

---

> ### Author Response · Authors · 2017-12-15
> **clarification**
>
> Thank you for your feedback, we are sorry that the semantic attention wasn’t explained clearly in the text. We indeed mean semantic attention as standard attention as you’ve guessed. By semantics, we meant word translation semantics (word f is translated to word e). Our assumption is based on the insights from (Koehn and Knowles, 2017) in which they computed match score between the most attended source word and the aligned word (produced by fast-align) and reported the match scores are higher than 70% for English->German, English<->Russian.
>
> By syntactic attention, we meant that when the model decided to translate a word f in the source side, we want to model also look at syntactic relations of the word f in an explicit way, such as the head word of f. We hope that this approach would project richer structured information from the source to the target.
>
> We have now included statistical significant test in the 1st revision. You are right that the gain of SA-NMT is not statistically significant when compared to flat (shared)-attention models. We also included significant test when compared against the flat (no-shared)-attention models. The updated results in Table (2) and (3) shows that sharing attention is beneficial for both NMT and grammar induction. Our results also suggest that there are two possible ways to get more structural information from the source side: using Structured Attention and sharing attention. The Flat attention well behaved in our experiments perhaps because the restriction of sharing attention makes it biases further to syntactic information, or dependency head in this case.
>
> We are sorry that the idea of sharing attention wasn’t well explained in our paper. We are working on the clarification and we will update it soon.
>
> In equation 3, we meant S

---

### Official Review · AnonReviewer2 · 2017-11-27
**Worthwhile ideas give useful insight into latent graph structures for NMT; somewhat weak results.**

**Rating:** 6
**Confidence:** 5

**Review:**

This paper describes a method to induce source-side dependency structures in service to neural machine translation. The idea of learning soft dependency arcs in tandem with an NMT objective is very similar to recent notions of self-attention (Vaswani et al., 2017, cited) or previous work on latent graph parsing for NMT (Hashimoto and Tsuruoka, 2017, cited). This paper introduces three innovations: (1) they pass the self-attention scores through a matrix-tree theorem transformation to produce marginals over tree-constrained head probabilities; (2) they explicitly specify how the dependencies are to be used, meaning that rather than simply attending over dependency representations with a separate attention, they select a soft word to attend to through the traditional method, and then attend to that word’s soft head (called Shared Attention in the paper); and (3) they gate when attention is used. I feel that the first two ideas are particularly interesting. Unfortunately, the results of the NMT experiments are not particularly compelling, with overall gains over baseline NMT being between 0.6 and 0.8 BLEU. However, they include a useful ablation study that shows fairly clearly that both ideas (1) and (2) contribute equally to their modest gains, and that without them (FA-NMT Shared=No in Table 2), there would be almost no gains at all. Interesting side-experiments investigate their accuracy as a dependency parser, with and without a hard constraint on the system’s latent dependency decisions.

This paper has some very good ideas, and asks questions that are very much worth asking. In particular, the question of whether a tree constraint is useful in self-attention is very worthwhile. Unfortunately, this is mostly a negative result, with gains over “flat attention” being relatively small. I also like the “Shared Attention” - it makes a lot of sense to say that if the “semantic” attention mechanism has picked a particular word, one should also attend to that word’s head; it is not something I would have thought of on my own. The paper is also marred by somewhat weak writing, with a number of disfluencies and awkward phrasings making it somewhat difficult to follow.

In terms of specific criticisms:

I found the motivation section to be somewhat weak. We need a better reason than morphology to want to do source-side dependency parsing. All published error analyses of strong NMT systems (Bentivogli et al, EMNLP 2016; Toral and Sanchez-Cartagena, EACL 2017; Isabelle et al, EMNLP 2017) have shown that morphology is a strength, not a weakness of these systems, and the sorts of head selection problems shown in Figure 1 are, in my experience, handled capably by existing LSTM-based systems.

The paper mentions “significant improvements” in only two places: the introduction and the conclusion. With BLEU score differences being so low, the authors should specify how statistical significance is measured; ideally using a technique that accounts for the variance of random restarts (i.e.: Clark et al, ACL 2011).
Equation (3): I couldn’t find the definition for H anywhere.

Sentence before Equation (5): I believe there is a typo here, “f takes z_i” should be “f takes u_t”.

First section of Section 3: please cite the previous work you are talking about in this sentence.

My understanding was that the dependency marginals in p(z_{i,j}=1|x,\phi) in Equation (11) are directly used as \beta_{i,j}. If I’m correct, that’s probably worth spelling out explicitly in Equation (11): \beta_{i,j} = p(z_{i,j}=1|x,\phi) = …

I don’t don’t feel like the clause between equations (17) and (18), “when sharing attention weights from the decoder with the encoder” is a good description of your clever “shared attention” idea. In general, I found this region of the paper, including these two equations and the text between them, very difficult to follow.

Section 4.4: It’s very very good that you compared to “flat attention”, but it’s too bad for everyone cheering for linguistically-informed syntax that the results weren’t better.

Table 5: I had a hard time understanding Table 5 and the corresponding discussion. What are “production percentages”?

Finally, it would have been interesting to include the FA system in the dependency accuracy experiment (Table 4), to see if it made a big difference there.

---

> ### Author Response · Authors · 2017-12-15
> **thanks for detailed comments**
>
> We would like to thank the reviewer for useful comments, and apologize for disfluencies in the original text.  We will absolutely prioritize clarity as we rework the writing in the final version of the paper.
>
> We agree with (Bentivogli et al, EMNLP 2016; Toral and Sanchez-Cartagena, EACL 2017) that NMT handles morphology better than phrase-based MT.  Isabelle et al does show that NMT can capture more morphology for French and English.  In our work, we choose German as a target language where long distance dependencies commonly occur.  We see that branching baselines perform measurably worse on the basic dependents of nouns and verbs in Table 4. We agree that it is reasonable to believe that existing NMT can handle the syntactic dependencies in an implicit manner, in our experiment (Figure 5), we show that if that information is available, the decoder prefers to use them, especially when predicting German verb. Additionally, when we designed our architectures, with the explicit goal of extracting interpretable structures, in part to compare the representations to prior linguistic knowledge of language. Both structured attention and gating norm to certain extent allow us to perform analysis on the task instead of training an additional classifier to probe the ability of the models in capturing linguistic phenomena.
>
> We have now included statistical significance tests in the 1st revision and we will make sure they are more clearly explained and pronounced in the final version. We have now also included significant tests when comparing against the flat (no-shared)-attention models. We truly appreciate that the reviewer noticed the shared attention mechanism we proposed even though we didn’t explain it well, something we are remedying by getting more eyes on our paper and isolating sources of confusion. We will try our best to make it more accessible in the next revision.
>
> You are correct in your understanding of the dependency marginals in Equation (11).  We have elaborated on this in the revision but are open to further suggestions.
>
> Regarding “production percentages” – While aggregate attachment numbers give a score to how well syntax is induced generally, they don’t give us insight into the grammar.  As a proxy for which grammatical rules the system has learned, we choose to analyze the frequency with which specific “head → child rules” were used by our model vs how often that rule exists in the grammar of the language.  For example, the three most common verb constructions are verb chains (VERB→ VERB), verb subj/obj (VERB→ NOUN) and verb’s being modified by an adverb (VERB→ ADV).  The gold column indicates how common these constructions are in the true data and the remaining columns show how often our systems believe these constructions exist.  We will need to spend more time in the next revision clarifying this demonstration.
>
> In equation 3, we meant S. We are sorry for the typo.

---

### Official Review · AnonReviewer1 · 2017-11-28
**Inducing grammars with and for NMT**

**Rating:** 5
**Confidence:** 4

**Review:**

This paper induces latent dependency syntax in the source side for NMT. Experiments are made in En-De and En-Ru.

The idea of imposing a non-projective dependency tree structure was proposed previously by Liu and Lapata (2017) and the structured attention model by Kim and Rush (2017). In light of this, I see very little novelty in this paper. The only novelty seems to be the gate that controls the amount of syntax needed for generating each target word. Seems thin for a ICLR paper.

Caption of Fig 1: "subject/object" are syntactic functions, not semantic roles.

I don't see how the German verb "orders" inflects with gender... Can you post the gold German sentence?

Sec 2 is poorly explained. What is z_t? Do you mean u_t instead? This is confusing.

Expressions (12) to (15) are essentially the same as in Liu and Lapata (2017), not original contributions of this paper.

Why is hard attention (sec 3.3) necessary? It's not differentiable and requires sampling for training. This basically spoils the main advantage of structured attention mechanisms as proposed by Kim and Rush (2017).

Experimentally, the gains are quite small compared to flat attention, which is disappiointing.

In table 3, it would be very helpful to display the English source.

Table 4 is confusing. The DA numbers (rightmost three columns) are for the 2016 or 2017 dataset?

Comparison with predicted parses by Spacy are by no means "gold" parses...

Minor comments:
- Sec 1: "... optimization techniques like Adam, Attention, ..." -> Attention is not an optimization technique, but part of a model
- Sec 1: "abilities not its representation" -> comma before "not"

---

> ### Author Response · Authors · 2017-12-15
> **updated grammar induction evaluation**
>
> Thank you for your insightful comments, we failed to properly convey the intention and novelty of the work.  The field of NLP has long prioritized structured model representations under the assumption that there are a fundamental property of language and therefore often a prerequisite for language tasks.  A naive reading of recent research in NMT seems to direct contradict this age old belief. Our goal here was to investigate if the success and seeming necessity of attention mechanisms is related to their ability to capture these structural/hierarchical properties of language.  The gating mechanisms and gains in BLEU score exist in service of this exploration, so so we agree are in and of themselves perhaps more minor contributions.  For example, you're absolutely correct that some of the basic components of our model are shared with previous work (Kim, 17 and Liu 17), but the shared attention and gating syntax are crucial in our models and their resulting analysis.  We will try to make this clearer in the final version.
>
> We originally ran our evaluation on Spacy’s outputs due to discrepancies between the MT tokens and tokenization and treebanks, but we have remedied this difference and updated all of our numbers to be evaluated against the Universal Dependencies treebanks.  The new results are in Table 3.
>
> There are two main results from Table 3: 1. Sharing attention appears to almost exclusively increase the model’s ability to capture syntax and 2. That structure attention generally outperforms flat attention if viewed through the same lense.  Almost secondary to these results is the fact that shared structured attention also benefits translation BLEU scores (updated with statistical significance).  This result does however hint that better modeling or inducing linguistic structure might further benefit translation performance.
>
> Finally, you are correct to question the inclusion of hard-attention.   While it is harmful for translation it appears to help grammar induction.  We hope that understanding this discrepancy and the possible (de-)coupling of the two metrics may lead to new results in future-work.  Maybe multiple syntactic analyses should be used as references instead of a single formalism?  In our experiments, hard-attention is deterministically computing by taking the max instead of sampling.
>
> We will rework our example and fix typos like z_t which you are correct should be u_t.

---

### Public Comment · ~Samuel_R._Bowman1 · 2017-11-29
**Neat!**

Do you have any examples of the structures learned with hard attention beside the tricky-to-read example in Figure 4?

---

> ### Author Response · Authors · 2017-12-15
> **visualizing trees**
>
> Thank you for your interest and encouragement!  We have been working to recompute numbers on UD for reviews (see below) so now that we have those we can try and build out better visualizations to share more standard dependency graphs.  We are sorry for the delay.

---

### Author Response · Authors · 2017-12-15
**Comments on structured Attention and sharing attention**

One of the aspects of this work we don’t feel we made sufficiently clear were the roles of structured and shared attention.
In this work, we explore structured attention (SA) as an explicit way to encode hierarchical structure of the source sentence. SA takes the global structure of a sentence into account as it needs to compute the partition function. For completeness, we also compare SA against self-attention (Flat attention) which only consider local dependencies between a word and its head. We evaluate the quality of induced tree on Universal Dependencies dataset. We observe that overall, SA models obtain better DA/UA scores, and SA are highly sensitive with the choice of architectures (sharing attention). Our original motivation for sharing attention is to help biasing SA to capture syntactic dependencies on the source side. This is reflected in both BLEU scores and DA/UA scores. Without sharing attention, DA/UA scores are considerably worse while the models still on par or outperform the NMT baselines. We find this result is exciting. It suggests that there should a better way to exploit SA for improving NMT as well as grammar induction.  Further, the fact that the choice of the target slide language changes these values hints at the different agreement syntax of languages so combining models may lead to further gains and syntactic learning in the future.
Finally, we also note that, for grammar induction, we are particularly interested in DE as long distance dependencies are more common in DE. The results show that structured attention are indeed superior to FA.  We leave these insights to future work to explore.

---

### Author Response · Authors · 2018-01-08
**Updated paper**

Dear reviewers,
We would like to let you know that we've updated our paper based on your valuable comments. Again, We thank you for your feedback!

---

### Decision · Program_Chairs · 2018-01-29
**ICLR 2018 Conference Acceptance Decision**

**Decision:**

Reject

**Comment:**

In this work reviewers use structured attention as a way to induce grammatical structure in NMT models. Reviewers liked th motivation of the work and found experiments mostly well done. However reviewers found the paper a bit difficult to follow, with several commenting that distinctions made between the different sub types of attention were not clear. Mainly the reviewers were not overwhelmed by the results of the work, saying that these gains, while clearly isolated to the use of structure were not significantly large. Additionally there were some concerns about the claimed novelty of the work, particularly compared to Liu and Lapata and other use of syntax in translation, and also which aspects were new or necessary.